# Light-driven peristaltic pumping by an actuating splay-bend strip

Klaudia Dradrach [1,2] ✉, Michał Zmyślony [1], Zixuan Deng[3], Arri Priimagi [3], John Biggins [1] ✉ & Piotr Wasylczyk [2]

Despite spectacular progress in microfluidics, small-scale liquid manipulation, with few exceptions, is still driven by external pumps and controlled by large-scale valves, increasing cost and size and limiting complexity. By contrast, optofluidics uses light to power, control and monitor liquid manipulation, potentially allowing for small, self-contained microfluidic devices. Here we demonstrate a soft light-propelled actuator made of liquid crystal gel that pumps microlitre volumes of water. The strip of actuating material serves as both a pump and a channel leading to an extremely simple microfluidic architecture that is both powered and controlled by light. The performance of the pump is well explained by a simple theoretical model in which the light-induced bending of the actuator competes with the liquid's surface tension. The theory highlights that effective pumping requires a threshold light intensity and strip width. The proposed system explores the benefits of shifting the complexity of microfluidic systems from the fabricated device to spatio-temporal control over stimulating light patterns.

Microfluidics is both a science and a technology[1]. On the one hand, it raises and answers fundamental questions about small-scale interactions between fluids and solids[2]. On the other hand, it enables microscale manipulation of fluids, with immediate applications in chemical analysis, biological assays, food safety and healthcare[3]. Inspiration is drawn from semiconductor-based electronic chips, but also from the entwined fluidic systems that pervade biological organisms. The flagship lab-on-a-chip technology aims at delivering miniature systems that integrate one or several laboratory functions on a single chip[4]. Complex chemical analysis or diagnostics can thereby be undertaken quickly, and on a small sample, permitting potentially revolutionary technological applications such as whole-genome sequencing from a single cell[5]. In the pursuit of these ends, most practical microfluidic devices consist of hair-sized channels on a glass or polymer plate, along with active or passive components to perform basic tasks such as mixing, dosing or pumping[6].

However, in stark contrast to the exponential complexification of electronic chips, commercial microfluidic chips remain rather simple. A recent review highlights several challenges[3]. Most importantly, microfluidic flows are typically powered and controlled externally, often with syringe pumps and pneumatic or magnetic valves. Consequently, although the chip itself is millimetre scale, the external devices make the whole setup orders of magnitude larger, limiting miniaturisation and portability[7]. The interface between the chip and these external devices is also a persistent problem: each join is a failure point, and the chip's perimeter is often crowded with external connections[8] preventing complexification. Secondly, most microfluidic chips have a fixed design for a fixed function, quite unlike their reprogrammable electronic counterparts, or a full-scale chemical laboratory. Thirdly, PDMS-based soft lithography, which is the academic standard for complex architectures, remains very challenging to use at an industrial scale[9]. These considerations motivate a search for new materials and approaches, which in turn raise new questions about fluid-solid interactions.

The integration of optics and microfluidics offers a compelling route forward. Much early work in this emerging field of optofluidics concentrated on using fluidics to reconfigure and tune optical functions[10]. However, there is also great appeal to using optics to

[1]Department of Engineering, University of Cambridge, Cambridge, United Kingdom. [2]Faculty of Physics, University of Warsaw, Warsaw, Poland. [3]Faculty of Engineering and Natural Sciences, Tampere University, Tampere, Finland. ✉e-mail: kd498@cam.ac.uk; jsb56@cam.ac.uk

control, power and monitor fluidic manipulations. Light is already routinely deployed to read out information from the microfluidic chips, often facilitated by the transparency of PDMS[1]. Going further, one can seek to illuminate a chip with structured light that both powers and controls fluidic tasks within it. This approach avoids external pumps and valves, and can leverage our exquisite ability to control light to flexibly steer complex reactions. Critically, light can penetrate through a chip's thickness dimension, removing the need for the internal control circuitry and circumventing the complexity bottleneck of crowded perimeters. Moreover, although initially, the optics may employ large external lasers and cameras, there is every reason to believe they could be miniaturised and integrated, enabling truly small devices.

Two basic approaches have emerged for implementing this vision. One option uses light to directly manipulate the fluid or its contents, for example by using optical tweezers to sort cells[11], or using light-induced gradients of surface tension to manoeuvre droplets[12,13]. Alternatively, light can be absorbed by a photo-responsive material, which mediates the interaction. For example, gels that reversibly swell in response to illumination have been used to create valves[14–16]. Similarly, liquid crystal (LC) polymers/networks containing azobenzene moieties undergo deformation in response to illumination stemming from azobenzene's photo-isomerisation[17]. These photodeformations have been used to create a reciprocating membrane micro-pump and micro-valve via photo-bending films[18,19], and to generate photo-actuating tubes, which propel liquid slugs using capillary forces[20,21]. However, photo-responsive gels are slow, taking several minutes to open a valve, and LC polymers, although faster, are limited to small deformations. Both also require illumination by potentially destructive UV light.

Liquid crystal elastomers (LCEs) are a promising alternative category of actuating material[22,23], that can offer large (~50%), reasonably fast and reversible strains in response to heat or light[24,25]. These materials are rubbery LC networks in which the molecules typically align into a nematic phase. Upon heating, LC alignment is disrupted, reflecting the conventional nematic-isotropic phase transition, and the LCE contracts substantially along the alignment direction, providing a muscle-like actuation, as shown in Fig. 1a. The pioneering demonstration of LCEs in microfluidics involved embedding an LCE microactuator into a channel to create a thermally switchable valve addressed by an adjacent Joule heater[26].

Optical control of LCEs can be achieved by incorporating azobenzene moieties so that the LCE actuates photochemically[27,28], but such an actuation cycle is slow and typically requires damaging UV light. Alternatively, optical actuation may be achieved by loading the LCE with absorbing dopants (e.g. dyes, carbon-based nanomaterials) and then heating photo-thermally[29–31]: this mechanism is simple to implement, tunable to any wavelength, and can operate quickly. Photothermal actuation is well established in LCE robotics/locomotion, but not in fluidic manipulation, possibly because early LCEs had rather high actuation temperatures (often ≥100 °C), which can damage or boil a fluid, and are hard to sustain in highly conducting fluidic environments. Fortunately, strategies have now emerged for lowering LCEs actuation temperature, even below body temperature[32–34], albeit with synthetic complexity. One simple strategy for lower temperature is to swell a highly cross-linked LC network with a low molecular weight LC, which plasticises the network to form a softer LC gel (LCG)[35,36]. Such LCGs do not actuate as dramatically as LCEs, but are attractive low-temperature actuators and have already proven well suited for underwater photothermal actuation[37].

Evolution has long harnessed muscular peristaltic to gently guide fluids and slurries through the digestive, lymphatic and reproductive systems. In engineering contexts such peristaltic pumping is valued for its valveless simplicity, low maintenance, and limited scope for contamination, and is commonly deployed at large scales to transport aggressive or rheologically complex fluids. These advantages also prevail at small scales, and, correspondingly, microfluidic peristaltic pumps have been demonstrated using linearly distributed piezoelectric[38,39], pneumatic[40–42], electrostatic[43] and electrothermal[44] actuators. However, all these architectures share the fabrication and control challenges that pervade microfluidics: multiple actuators must be embedded along a channel, and supplied with power and control lines which, in turn, exit at the perimeter. Promisingly, several previous papers have obtained peristaltic-like waves of actuation in LCE structures by scanning a stimulating light along their length, allowing crawling on land[45] and underwater[37] and also underwater swimming[29], all without physical control lines.

Here we combine these ideas to create a pump which transports fluids via a peristaltic wave of photo-deformation along an LCG strip. The strip sits atop a stiff platform, and a scanning laser spot generates and propagates a fluid-filled bump between film and plate. This

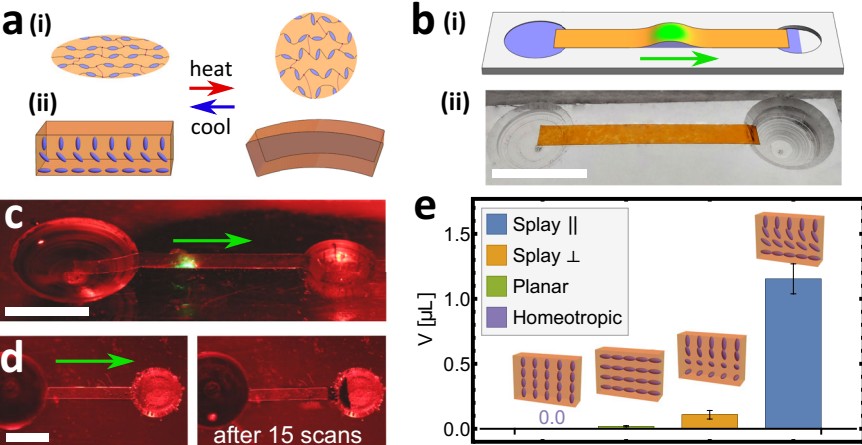

**Fig. 1 | The design and demonstration of an LCG photo-peristaltic pump. a** (i) Schematics of an LCE showing contraction along the direction of the alignment. (ii) Bending of an LCG sheet with splay-bend director caused by the bottom side contracting and the top side expanding on heating. **b** (i) Schematic picture of experimental system with the arrow denoting the scanning and fluid transport direction, and (ii) a picture of an unstimulated system. **c** Photograph of the experimental setup, including the light-induced peristaltic bump. **d** Photographs before and after 15 scans of the laser light to demonstrate the transport of water, which was dyed black for better contrast. **e** Comparison of amounts of liquid transported in a single scan for four different LC alignments. All strips had widths of 1.7 mm and were illuminated with a peak light intensity of 12.8 W/cm². Scale bars are all 1 cm.

optofluidic design dispenses not only with physical control lines but also with channels themselves. In contrast, previous demonstrations of LC-actuators for microfluidics have either embedded actuators into channel-based chips for valving[18,26] or pumping[19], or even constructed the channels themselves out of LC material[20,21]. Our design leads to an open system that is easy to fabricate and can support reprogramming by changing the stimulation pattern. Furthermore, pumping indeed operates via an unusual solid-fluid interaction: a competition between active bend and surface tension. We capture this competition in a simple quasistatic theory which reveals a rich nonlinear dependence of bump shape on the stimulation and design parameters, and a threshold illumination for pumping.

## Results

### Design of the photo-peristaltic pump

As discussed, photothermal actuation of LCGs offers a promising basis for optically driven peristalsis. We thus initially fabricated strips of LCG with a longitudinal nematic alignment and doped them with an absorbing red dye. As detailed in the methods, fabrication is via synthesis from low-molar-mass LC (acrylate homopolymerisation), in which a nematic mixture of reactive mesogens is dissolved in non-reactive nematogenic solvent (5CB), then this melt is flooded into a 50 μm thick cross-linking cell and finally UV cured. The result is a comparatively stiff nematic network (LCN) swollen with low molecular weight nematogen, giving a softer and more responsive LCG[37]. Alignment of the melt is achieved via surface anchoring on the inner surfaces of the cell, and then locked in by cross-linking[45,46]. As shown in the SI, a planar-aligned strip has Young's moduli of $E_\parallel \approx 40$ MPa and $E_\perp \approx 8$ MPa (Fig. S3), and contracts on heating at 0.1 %K$^{-1}$ (Fig. S4).

The next task is to identify a spatial and temporal pattern of contraction that pumps effectively. In biology, gut peristalsis is achieved via contractile muscles that wrap azimuthally around the intestine and squeeze it to reduce its bore. A strongly biomimetic approach would thus be to create an LCG tube with an azimuthal director. However, such structures are challenging to fabricate at milli- or microscale (c.f. the fabrication in ref. 20), requires illumination from all sides to operate, and would deliver disappointing occlusion at the available strains. In contrast, flat LCG sheets offer a more attractive approach: they are straightforward to fabricate, can be encoded with very sophisticated patterns of contraction via surface anchoring[46], and could naturally form a layer in a planar microfluidic device and be conveniently actuated by the light incident from above. These considerations lead us to the design in Fig. 1b, where a freestanding flat strip of LCG sits between two wells on a flat, stiff polymeric plate, lubricated by a very thin fluid layer. Actuation by a scanning light spot then creates and propagates a fluid-filled bump which transports fluid between the wells.

In order to implement this idea, we must encode a pattern of contraction into the strip that will sculpt a bump on local stimulation. Longitudinal alignment is clearly inappropriate, as it would lead to a shortening of the stimulated region, which would thus be in tension and remain flat. Instead, we create a strip in which the director rotates through the thickness from longitudinal on the lower surface to homeotropic on top. As shown in Fig. 1a, stimulation of such a splay-bend sheet leads to longitudinal contraction on the bottom and extension on the top, causing bend[22,47], much like a bimetallic strip[48]. Such bending delivers large curvatures even with modest strains, and localised stimulation by a scanning light will cause a region of spontaneous bend to create a fluid-containing bump.

An initial demonstration is shown in Fig. 1c, using a 1.7 mm wide splay-bend strip, spanning between wells separated by ~1.5 cm and with each end overhanging. Actuation in a small region by a green laser spot (of size comparable to the strip width) leads to the formation of a localised bump. Furthermore, scanning along the strip from one well

to the other causes the bump to follow the light and transport fluid between the wells (Fig. 1d, Supplementary Movie 1).

All of our results use a scanning speed of 1.2 mm/s, which, empirically, is the fastest that allows full LCG deformation, rather than being limited by rate-dependent effects such as viscoelasticity. Between pumping scans, the laser returns via a much faster backwards scan, which delivers little energy and no appreciable deformation. The magnitude of the forward quasistatic deformation is then determined by the light-induced temperature profile of the LCG, and hence by the optical energy delivered. For this spot size, peak intensities of around 13 W/cm$^2$ are required for effective pumping. We also directly observe the resultant temperature using a thermal imaging camera. A trace of the temperature at a fixed location on the strip over multiple scans (Fig. S2) reveals that, for a peak light intensity of 13 W/cm$^2$, the temperature rises to 47.5 °C during scan, and falls to 26 °C between. Successful pumping with such a modest temperature validates the choice of a swollen LCG actuator.

Our design contends that the propagating bend wave of a splay-bend LCG strip will be particularly effective. However, in principle, peristalsis may be driven by any travelling deformation in intimate contact with a fluid. To examine this possibility, we also constructed pumps based on three strips with alternative deformation: longitudinal planar alignment (giving longitudinal contraction), homeotropic alignment (longitudinal expansion), and perpendicular splay-bend alignment (bend in the width direction). As anticipated, both longitudinal and homeotropic are indeed extremely ineffective with no appreciable bump forming at all. Naively one might expect the perpendicular splay-bend strip to pump effectively by creating a bump via perpendicular bending. However, such transverse curvature is actually suppressed, as it would require a doubly-curved bump that carries Gauss Curvature and hence, via the *theorema egregium*, stretching of the strip. Consequently, as seen in Fig. 1e, the original design is markedly the most efficacious.

### Shape equation

To understand and analyse the pump's action, we develop a 1D model of the propagating deformation's shape. The spontaneous bend is localised around the illumination, which, for a 1D model, we take to only vary in the length direction. Although this is a simplification for a circular laser spot, we were able to implement it experimentally by first uniformly expanding the beam and then contracting in the length direction with a cylindrical lens, so that the longitudinal FWHM is equal to 2.7 mm while the transverse FWHM of 4.55 mm is significantly wider than the strip itself. In order to maintain the overall rate of energy delivery, the peak intensities ranged between 1.2 and 4.8 W/cm$^2$. Side views of the resultant bump are shown in Fig. 2c and Supplementary Movie 2, confirming that this change does not qualitatively change the pump's action.

Uniform heating causes the strip to roll with uniform curvature $\bar{\kappa}$ (Fig. S5), which is an increasing function of temperature (Fig. S6), and consistent in magnitude with previous theory[22]. Localised stimulation of an unconstrained strip would then form a high amplitude blunted V-shape, as illustrated in Fig. 2a and shown experimentally in Fig. S7. Evidently the fluid must restrict this actuation, leading to the observed finite bump. Dynamic effects from viscosity or inertia may be ruled out due to the low scanning speed, as may gravity due to the small size. Instead, we hypothesise that the surface tension $\gamma$ of the liquid-air interfaces at the sides of the strip (blue area in Fig. 2b) is the main competing factor; deviating from the strip's preferred curvature costs elastic energy but reduces the liquid-air interface.

To capture this competition, we consider a strip of width $w$ that contains a localised bump described by the height function $y(x)$, as shown in Fig. 2b. Given the bump gradient $y'(x)$ is shallow in all our

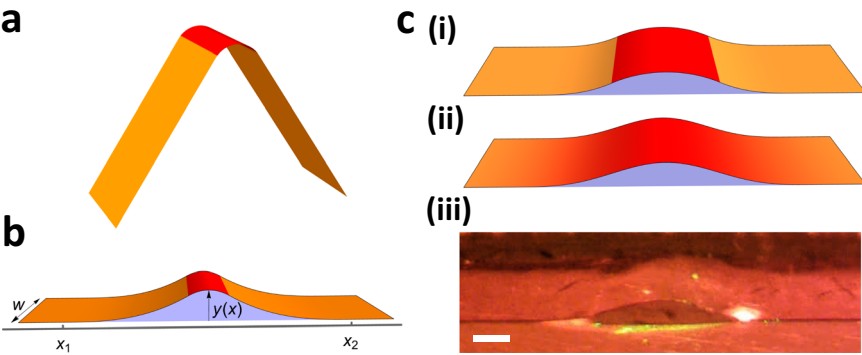

**Fig. 2 | Modelling the shape of the actuated peristaltic pump. a** Blunted V-shape deformation minimising the elastic energy in the absence of liquid. **b** Schematic picture of the pump, showing the quantities used in the development of the theoretical model. **c** Comparison of deformation profiles of 3.7 mm wide strips predicted by theory using (i) rectangular and (ii) Gaussian preferred curvature profiles with $l = 2.7$ mm and $\kappa_m^R = 0.8$ mm$^{-1}$ for the rectangular stimulus and $\kappa_m^G = 1$ mm$^{-1}$ for the Gaussian, and (iii) a snap-shot from the experiment in which the beam has FWHM matching $l$ and peak intensity of $I_m = 3.55$ W/cm$^2$. All the images in the panel **c** are in scale with the scale bar of length 0.5 mm. Throughout the figure the red marks the stimulated part of the strip, while the orange the unstimulated.

experiments, we may ascribe a total energy

$$E = \frac{Dw}{2}\int_0^L (y''(x) - \bar{\kappa}(x))^2 \, dx + 2\gamma \int_0^L y(x) \, dx. \qquad (1)$$

The second term above accounts for the surface energy of the liquid-air interfaces on both sides, while the first term is the bending energy incurred when the strip's actual curvature $y''(x)$ deviates from the preferred one $\bar{\kappa}(x)$ caused by illumination. Bending energy scales with flexural rigidity which, for a simple strip, can be computed from Young's modulus $E_y$, thickness $t$ and Poisson ratio $\nu \approx \frac{1}{2}$ as $D \equiv E_y t^3/(12(1 - \nu^2))$. The appropriate $E_y$ for a splay-bend strip is a subtle question, as the material has different moduli parallel and perpendicular to the director, and both are active during bending. Accordingly, we approximate $E_y \approx 30$ MPa, the figure measured during longitudinal stretching of a splay-bend sample itself, during which stretching parallel and perpendicular to the director are both active.

To deduce the shape, we minimise the energy variationally over $y(x)$, leading to an Euler-Lagrange equation

$$y^{(4)}(x) - \bar{\kappa}''(x) = -\frac{\gamma}{Dw}, \qquad (2)$$

and corresponding bump solution

$$y_b(x) = c_0 + c_1 x + c_2 x^2 + c_3 x^3 - \frac{x^4 \gamma}{12 Dw} + \bar{y}(x), \qquad (3)$$

where $\bar{y}(x)$ is the preferred shape of the strip, given by $\bar{y}''(x) = \bar{\kappa}(x)$, and the $c_i$ are constants of integration. At first sight, this form appears problematic as it diverges for $x \to \pm \infty$. The key additional ingredient is that the strip cannot penetrate the substrate, so minimisation is subject to the constraint $y(x) \geq 0$. As seen in Fig. 2b, we thus have a finite bump region where contact is lost (following $y_b(x)$ above) that connects at finite locations $x_1$ and $x_2$ to contacting material ($y(x) = 0$) in the flanks. The contacts must have $y_b = y_b' = 0$ for continuity and finite curvature, which fixes the $c_i$. Finally, minimising the energy over $x_1$ and $x_2$ gives the additional boundary condition $y_b'' = 0$ fixing the values of $x_1$ and $x_2$ themselves.

A simple form of preferred curvature is a rectangle function of magnitude $-\kappa_m$ and length $l$, defined (in terms of the symmetric unit rectangle function $\Pi(x)$) as $\bar{\kappa}^R(x) = -\kappa_m \Pi(x/l)$ (Fig. S8). In this case, we

may find the complete solution

$$y_b^R(x) = \frac{3}{32}(\kappa_m l)^{4/3}\lambda + \frac{3}{4}\frac{(\kappa_m l)^{2/3}}{\lambda}x^2 + \frac{x^4}{2\lambda^3} - \kappa_m l^2 \begin{cases} \frac{1}{8} + \frac{x^2}{2l^2} & |x| \leq l/2, \\ \frac{|x|}{2l} & |x| > l/2, \end{cases} \qquad (4)$$

$$x_2 = -x_1 = \frac{1}{2}(\kappa_m l)^{1/3}\lambda, \qquad (5)$$

where, for conciseness, we have above introduced the length-scale $\lambda = (6Dw/\gamma)^{1/3}$, which emerges naturally from balancing bending and surface tension. Despite the simplicity of our theory, this shape has a sophisticated and nonlinear dependence on the stimulus, suggesting complex behaviour.

To verify that the theory captures the essence of the pump, we evaluate the predicted shape for a strip of the width of 3.7 mm, $\gamma = 0.72$ mN/m for a water-air interface, and a stimulus with a length matching the FWHM of the experimental beam. As shown in Fig. 2c, this substitution leads to a realistic shape that compares well with a similar experiment provided we take a peak curvature corresponding (Fig. S6) to a sensible peak temperature of around 55 °C. A more realistic model would be a Gaussian preferred curvature profile, $\bar{\kappa}^G(x) = -\kappa_m \exp\left(-4\ln(2)x^2/l^2\right)$, in which $l$ is the FWHM of the distribution and $\kappa_m$ is the maximal preferred curvature (Fig. S8). Although this profile does not admit an analytical solution, minimisation can easily be done numerically (Methods) and yields a very similar shape as presented in Fig. 2c (ii). One difference is that the Gaussian requires larger values of peak preferred curvature for deformations of similar amplitudes, as it decays away from the centre; correspondingly, the deformation in Fig. 2c (ii) has a peak temperature of around 65 °C.

In our calculations, we have omitted gravity. As always in a fluid, the competition between gravity and surface tension is captured by the capillary length (3 mm for water), with surface tension controlling below this scale, and gravity above. Given the bump amplitude is a magnitude smaller, we indeed expect a surface tension-dominated regime. We confirm this conclusion by numerically including gravity (Fig. S9), and also by observing that the strip can transport a fluid bolus even if the whole apparatus is inverted (Supplementary Movie 3). However, these considerations do not apply to the reservoirs, which have diameters comparable to the capillary length, and rely on gravity to contain the fluid. Correspondingly, inflow and outflow involve elasticity, capillarity and gravity. The strip's end extends over the reservoir and we observe (Supplementary Movie 2) that it remains on the water-air interface while it (un)bends during (out)inflow, and that

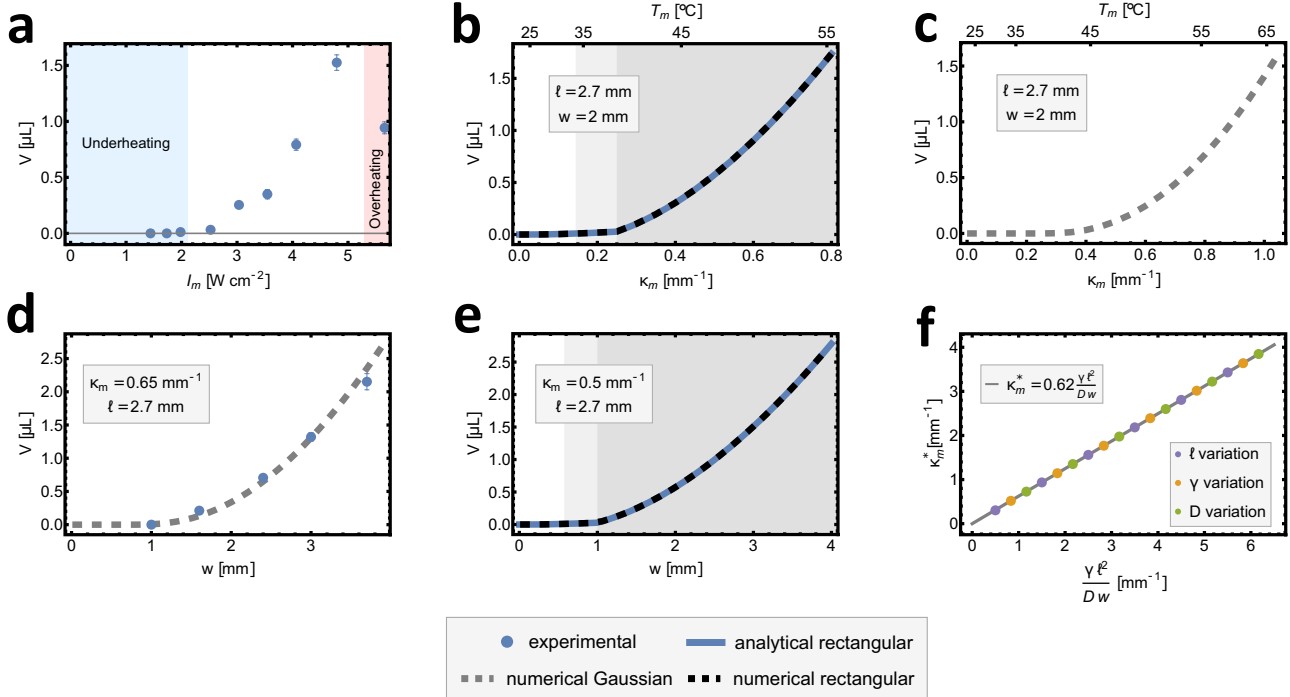

**Fig. 3 | Volume curves showing the thresholds for both the experiment and the theory. a–c** Volume transported in a single scan depending on the maximal intensity of the laser beam or, in the case of theory, the maximal curvature. A clear threshold behaviour can be seen in experimental and theoretical plots. The temperature axis corresponds to the peak temperatures calculated from curvature-temperature data presented in Fig. S6. **d, e** A similar threshold was observed (**d**) and predicted (**d** Gaussian, **e** rectangular) when the width of the strip was varied. The experimental beam in **d** had peak intensity of $I_m = 3.55$ W/cm², and the values of $\kappa_m$ were calibrated using **a–c** to give the correct volume for $w = 2$ mm. **f** Calculated threshold curvatures for the Gaussian stimulus follow the same scaling law as predicted by analytical calculations for the rectangular stimulus. Throughout colours of the background of the theoretical plots correspond to different response regimes, with white being the weak response, light grey intermediate and darker grey strong.

both processes are effective at various reservoir fill levels. Our interpretation of the motion is that the comparatively large reservoir area reduces capillary forces during inflow processes, allowing larger amplitude bending that deforms the air-water interface via normal meniscus physics, and can easily capture enough water to fill the bump profile. Similarly, during outflow, the transported fluid rapidly enters the reservoir upon contact to eliminate the higher surface energy of the sides in favour of a low-energy deformed meniscus.

## Pumping thresholds

We next explore experimentally and theoretically how pumping performance depends on the design parameters. Firstly, we measure how the transported volume changes with laser intensity. To do this, we use a strip of a width of 2 mm, stimulated by a laser beam with a longitudinal FWHM of 2.7 mm, and varied the intensity of the beam by directly using controls on the laser. As seen in Fig. 3a, we observe that liquid transport only occurs above a threshold intensity and then rises sharply before again falling at high intensity due to evaporation. This final consideration provides a second motivation for reducing the actuation temperature by swelling the LCG, as it not only avoids damage to biological fluids but also prevents evaporation.

As a first step to understanding this threshold theoretically, we find numerically the bump profile for a Gaussian stimulus with $l = 2.7$ mm for a range of values of $\kappa_m$ corresponding to the expected temperatures across the experimental intensity range. We can then evaluate the transported volume per scan as $V = w \int y(x)dx$, leading to the curve in Fig. 3c, which indeed shows a threshold magnitude of the stimulus and a good agreement with the experimentally observed volumes, despite a complete lack of fitting parameters. The experimental and theoretical curves (Fig. 3a and c) can then be used to calibrate the appropriate $\kappa_m$ for a given experimental intensity.

To understand the threshold further, we return to the rectangular stimulus profile, for which we may analytically calculate the volume as

$$V^R = \frac{1}{120} w \kappa_m l \left( 3(\kappa_m l)^{2/3} \lambda^2 - 5l^2 \right). \quad (6)$$

This result evidently encodes a threshold, as for sufficiently low $\kappa_m$ the volume actually goes negative. Indeed, looking more carefully, we see that, even before reaching this point, at

$$\kappa_m^* = \frac{4^3}{3^3} \frac{l^2}{\lambda^3} = \frac{2^5}{3^4} \frac{l^2 \gamma}{Dw}, \quad (7)$$

$y_b^R(0)$ goes negative, implying that the midpoint of the strip penetrates the substrate. A more careful treatment (Supplementary Information) reveals that our original formula for $y_b^R$ is valid for $\kappa_m \geq \kappa_m^*$ (the strong response regime), while below it, there are two different regimes with a contact at the origin, as seen in Supplementary Movie 4 and corresponding plot in Fig. 3b. More precisely, in the intermediate regime, the contact is only at a point, while in the weak response regime, the contact is over an extended width and the bumps become localised to the edges of the rectangle. These response regimes are associated with a step-like nature of the rectangular stimulus, so they are not observed in the experiment, and, as seen in Fig. 3b, they would not pump efficiently owing to their small volume. We thus conclude that the onset of high curvature regime $\kappa_m \geq \kappa_m^*$ is the appropriate measure for the pumping threshold.

Conversely, repeating our theoretical calculation but keeping $\kappa_m$ constant and varying the width leads to volume curves with a clear threshold width below which the pumping does not occur (Fig. 3e− rectangular, and 3d−Gaussian). Reducing width actually has two

strong effects: it reduces the volume for a given height and also reduces the strength of the elasticity but not surface tension, leading to a reduction in height itself and eventually preventing pumping altogether. Since the threshold width would depend on $\kappa_m$, one can always make a narrow strip pump by raising the degree of actuation. However, in practice, a too strong stimulus leads to evaporation leading to maximum curvature and therefore, via Eq. (6), a minimum width

$$w_{\min} = \frac{2^5}{3^4} \frac{l^2 \gamma}{D \kappa_m^{\max}}. \qquad (8)$$

Interestingly, the similarity between the width and preferred curvature is even more fundamental, as it can be shown from Eq. (2) that the volume does not depend on either one of these variables independently, but rather on their product (Supplementary Information).

To observe this threshold width experimentally, we measure transported volume for strip widths between 1 and 3.7 mm, with a fixed peak intensity and stimulation length. The resulting volume width plot is presented in Fig. 3d and confirms that the pumping only occurs over a threshold width. Again, agreement with the Gaussian calculation is strikingly good, despite no fitting parameters.

Although our threshold formula stems from the theoretical calculations for a rectangular stimulus, we expect the scaling behaviour to be universal, giving

$$\kappa^* \sim \frac{l^2 \gamma}{D w}. \qquad (9)$$

To test this, we compute the threshold curvature for a Gaussian stimulation profile (defined by $V^G = V^{R^*}$) for a range of values $D$, $\gamma$ and $l$. We find a perfect collapse onto the above relationship with a constant of proportionality of 0.62 (Fig. 3f) as compared to 0.40 for a rectangular stimulus.

### Pumping optimisation

Finally, we consider changing the stimulus length $l$ for a given strip and preferred curvature. Maximising the analytic volume (Eq. (6)) reveals an optimal stimulus length and corresponding maximal volume,

$$l_{\mathrm{opt}} = 3^{-3/4} \sqrt{\kappa_m \lambda^3} = \sqrt{\frac{2 D w \kappa_m}{\sqrt{3}\gamma}}, \quad V_{\max} = \frac{1}{270 \times \sqrt[4]{3}} w \kappa_m^{5/2} \lambda^{9/2}, \quad (10)$$

which for characteristic values ($w = 2\,\mathrm{mm}$, $\kappa_m = 0.7\,\mathrm{mm}^{-1}$) would be $V_{\max} = 1.3\,\mu\mathrm{L}$ and $l_{\mathrm{opt}} = 3\,\mathrm{mm}$, indicating that our previous experiments were close to optimal. Plots of the volume length curve for both a rectangular stimulus (Fig. 4b) and a Gaussian stimulus (Fig. 4a) confirm this result. Interestingly, Gaussian stimulation shows no pumping at large lengths, while rectangular stimulation shows small but persistent pumping via small bumps at the stimulus' discontinuities (Supplementary Movie 5).

The discrepancy between the responses to Gaussian and rectangular stimuli clarifies why there is an optimal length and, indeed, the true pumping mechanism: a complete bump contains equal and opposite amounts of positive and negative curvature, while the local stimulation only promotes the negative. If the entire strip was homogeneously stimulated, there would thus be no advantage to a bump. Therefore, the bump only forms because the negative curvature aligns with the stimulus, while the positive curvature is beyond it, providing a net energetic advantage. Indeed, Eq. (2) reveals that the second derivative of preferred curvature is critical to forming a bump, explaining why a rectangle function's edges can always form a small bump while a diffuse Gaussian cannot.

A corresponding experiment was conducted by varying the spacing between the cylindrical lens and the strip so that the longitudinal

beam length ranged between 1.3 and 3.2 mm, while the total laser power was appropriately adjusted to keep peak intensity constant. Data for this experiment are shown in Fig. 4a. The experimental data confirm optimum length is close to that predicted. Interestingly, pumped volumes at small stimulus lengths fall off faster than the theoretical prediction, likely owing to thermal broadening which becomes significant when the stimulus is very narrow.

The optimised length also allows us to estimate the maximum achievable flow rate as $\dot{V} = V_{\max}\, v/L$, where $v$ is scanning speed and $L$ is strip length. The strip from Fig. 4b has $L = 20\,\mathrm{mm}$ giving a maximal flow of 4.8 μL/min.

### Versatility of the LCG peristaltic pump

Having understood the pump's basic mechanics, we finish by briefly demonstrating the versatility enabled by its design. Firstly, control with patterned light incident from the third dimension allows pumping patterns to be driven in the same strip, making the system reprogrammable. As a simple demonstration, reversing the scanning reverses the pumping direction (Fig. 5a and Supplementary Movie 6). Further extensions (requiring more than one light spot) could include multiple boluses in series, or rhythmic mixing actions under the strip similar to segmentation in the gut. Secondly, the strip can be easily moved to connect different reservoirs (Fig. 5b), making the system reconfigurable and, after cleaning, reusable. Thirdly, peristalsis is also particularly suited to complex fluids/slurries, and our strip can easily pump a complex, multi-component mixture (Fig. 5c). Fourthly, one may use multiple strips (joined or individual) to connect multiple reservoirs and pump between any pair. In Fig. 5d and Supplementary Movie 7 we demonstrate a cross-shaped sheet connecting four wells and capable of pumping between any two, although future optimisation is required to reliably navigate the centre.

### Discussion

We have presented an effective optofluidic method for transporting micro-portions of liquid over a solid surface by using peristaltic deformations of an LCG strip that are controlled and powered by light. Peristalsis transports a ~1 μL bump of fluid at a respectable microfluidic velocity of 1.2 mm/s, and is driven by modest-intensity green light, which avoids overheating or exposure to damaging UV radiation. Our study thus confirms that photothermal strains in LCGs are sufficiently large, fast and low-temperature for microfluidics, even with delicate biological fluids.

The pump is strikingly simple in conception and fabrication: a splay-bend LCG strip rests on a polymer platform, and is stimulated by scanning illumination. This design avoids all valves, external mechanical connections, or complex small-scale fabrication. The complete lack of contact between fluid and external machinery allows for handling of delicate fluids, potentially while sealed in a controlled atmosphere. Furthermore, one may easily adjust strip placement or stimulation pattern, both during and between tasks. Such adaptability could be valuable for rapid prototyping, in-field applications, or when fluidic function must be added post-hoc, e.g. between cultures in a Petri dish. More ambitiously, adaptability points towards a universal "lab-on-a-chip" that, like a full-scale laboratory or computer chip, can execute many different functions sequentially. Overall, we believe this approach of simple actuating structures stimulated by complex structured light is an exciting avenue for optofluidics, reminiscent of the dynamism and flexibility of electro-wetting droplet approaches[49,50].

Interestingly, although there are many demonstrations of LC-actuators in robotics, including aquatic locomotion[29,37,51,52], there are comparatively few previous demonstrations of fluid manipulation[18–21,26], which all, unlike here, proceed by embedding the actuators in or around channels. Our work also differs in the use of photothermal actuation (avoiding UV) and in the use of a spatial director profile to achieve a desired actuation—both well established in LCE actuation but not for

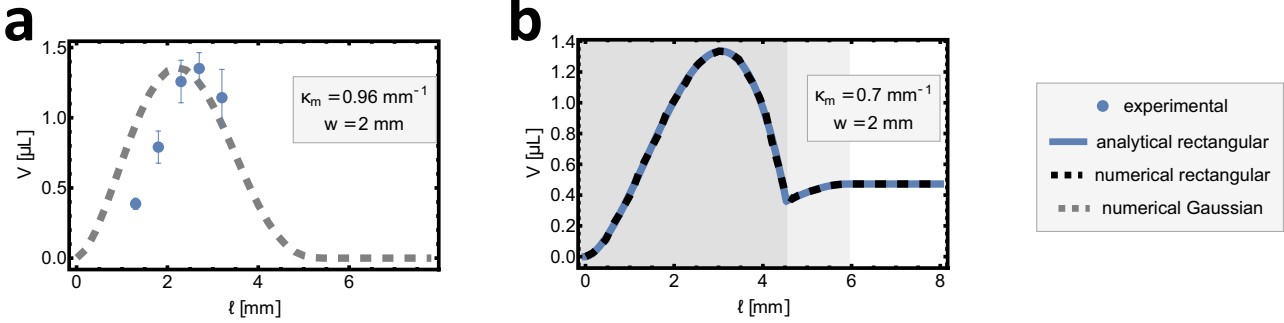

**Fig. 4 | Volume optimisation for different sizes of stimuli profiles at fixed peak intensity. a** Comparison of the volumes measured experimentally using peak intensity of $I_m = 4.61$ W/cm², and volumes predicted using Gaussian stimulus. **b** Theoretical plot for a rectangular stimulus profile, in which the background colouring follows the same convention as in Fig. 3.

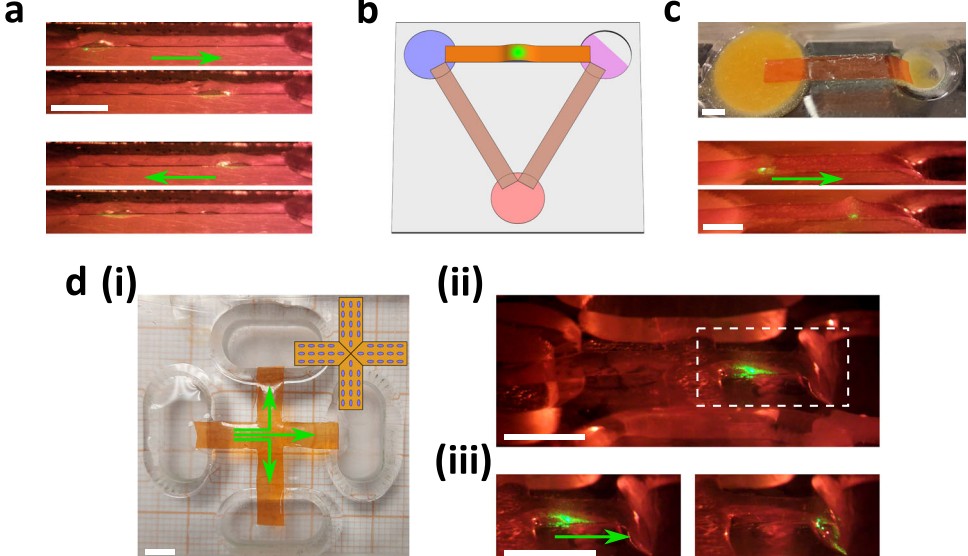

**Fig. 5 | Versatility of the LCG peristaltic pump. a** Reversibility: snap-shots showing pumping first to the right side, and then to the left side, taken from Supplementary Movie 6. **b** Reconfigurability: a simple pick-and-place approach can be used for pumping liquids between a number of reservoirs using a single actuating strip. **c** Complex fluid pumping: the pump can not only pump simple liquids such as water but also more complex ones like pumpkin soup. **d** Design complexification: (i) A cross-shaped actuator can connect four containers, and a peristaltic deformation can pass in any route via control over the light path. The inset shows the alignment at the bottom surface, with the top surface being homeotropic everywhere. (ii) A zoomed-out snap-shot of the pump showing a bolus of fluid being transported in one arm of the cross. (iii) Two snap-shots of the pump showing the transport of a portion of water and its depositing. The white marks the area of the zoomed-in pictures. The scale bars are all 5 mm.

fluid manipulation. Given these differences, it is perhaps unsurprising that our pump is governed by different mechanics, competition between active bend and surface tension, which we have captured in a simple analytic model.

Naturally, the LCG pump has limitations, which will inform future designs. Pumping rate for a single bolus is inversely proportional to strip length, making long pumps slow. Fortunately, this issue could be easily circumvented by having multiple boluses progressing simultaneously. Indeed, if boluses are placed end to end, a length-independent pumping rate of $\dot{V} \sim V_{max} v / l_{opt}$ could be achieved, which evaluates to 18 µL/min for Fig. 4b's parameters. More importantly, surface tension becomes increasingly important at small scales, leading to a minimal strip width below which effective pumping ceases, which limits miniaturisation. The bolus's open sides also allow evaporation and fluid leakage. These considerations suggest a promising future direction—an entire 2D sheet of actuating material (rather than a 1D strip) stimulated via a light spot to generate and manoeuvre a fluid-containing blister. This design eliminates the exposed bolus surface, and allows for

complicated 2D trajectories. More subtly, our 1D bump depends on bending deformations, driven by through-thickness actuation variation, which are inevitably fairly weak. However, Gauss's *theorema egregium* dictates that a doubly curved 2D blister requires an in-plane variation to form, and then cannot be defeated without energetically costly stretch. Such variation can be achieved via patterns of alignment[53], or patterns of actuation strength[31,54], but do require geometrically large strains that are only achievable with LCEs. As an example of the power of such *metric-mechanics*, LCE sheets programmed to rise into cones on heating/illumination[55] are capable of lifting thousands of times their own weight[56]. A 2D approach thus promises increased mechanical strength, opening the door to miniaturisation and making fuller use of the stimulation.

## Methods
### Materials
The LCG mixture was composed of standard LC reagents [(molar %): 78.5% of monofunctional reactive mesogen ST03866

(4-methoxybenzoic acid 4-(6-acryloyloxyhexyloxy)phenyl ester), 20% of bifunctional reactive mesogen ST00975 (1,4-Bis-[4-(6-acryloyloxyhexyloxy)benzoyloxy]-2-methylbenzene), 1% of photoinitiator (2,2-Dimethoxy-1,2-diphenylethan-1-one), 0.5% of red dye–Disperse Red 1 acrylate (DR1; N-Ethyl-N-(2-acryloxyethyl)-4-(4-nitrophenylazo)aniline) augmented by 5CB (4-Cyano-4′-pentylbiphenyl) (30% of the weight of LCN melt). Mono- and bifunctional reactive mesogens serve as monomer and cross-linker in the nematic melt, respectively. Monomer and crosslinker were obtained from Synthon; photoinitiator, Disperse Red 1 acrylate and 5CB from Sigma–Aldrich.

### Fabrication of light-driven pumps

Clean glass slides with a usable surface of ~2.5 × 2.5 cm were spin-coated with either poly(vinyl alcohol) (PVA)-water solution, or with polyimide derivative (PI) (N-Methyl-2-Pyrrolidone solution). Thin layers of PVA were then rubbed with the use of rubbing cloth for planar alignment, and thin layers of PI provide homeotropic alignment. Then, one PVA-coated and one PI-coated glass slide were assembled into 50 μm thick glass cell. In order to prepare an LCG film with unidirectional (planar) alignment, two glass slides with mechanically rubbed PVA layers were assembled together. Similarly, for samples with vertical alignment, two PI-spun coat glass slides were assembled into 50 μm thick glass cell.

The weighted compounds (monomer, cross-linker, photoinitiator, DR1, 5CB) were melted and mixed in a glass vial at 80 °C on a hot plate under magnetic stirring. Next, the glass cells were heated up to 80 °C and filled with the molten LC mixture by capillary forces. After cooling down to 40 °C, UV LEDs (peak wavelength: 365–380 nm) were used to crosslink the filled cells (for 5 min), and the freestanding films were obtained (schematic view of LCG is presented in Fig. S1). All the pumps were cut manually from the films.

### Characterisation of LCG actuators

Stress-strain characteristics were evaluated at a stretching speed 0.01 mm/s until fracture for LCG strips (size: 13 × 3 × 0.05 mm) using a custom-built tensile testing machine. The force sensor KD34s (ME-SYSTEME) with a force limit of 5 N was used. A LabVIEW 2017 software was adopted to detect the force-strain relations. Samples were secured with paper grips to prevent slipping. Tensile tests for planar alignment (stretching along and perpendicularly to the alignment), homeotropic and splay alignment were made (Fig. S3).

The thermal contraction of an LCG strip was measured with the use of a polarising optical microscope (Zeiss Axio Imager A1 Polarized Light Microscope), equipped with a Linkam stage (THMS350EV-2) by comparing samples' length change in elevated temperatures. Millimetre strips cut manually from LCG sheets were placed on a glass slide. Contraction parallel to the director in the sample during the heating and cooling cycle (10 °C/min) is presented in Fig. S4.

### Actuation of light-driven pumps

A continuous wave green 532 nm laser (Verdi V-5, Coherent) was used to power and control the pumps. The optical setup was equipped with basic optical elements like mirrors, lenses, and other components. The laser beam was steered with a mirror mounted on a galvanometer optical scanner (Cambridge Technology, 6240H), driven with a ramp signal from a waveform and function generator (Twintex, TFG-3620E, Arbitrary Waveform DDS Function Generator, 20 MHz). For measurements with different stimulus' lengths, two lenses were used to expand a beam (LA1131, $f = 50$ mm and LA1433, $f = 150$ mm, Thorlabs), which was later directed to the cylindrical lens (H1054) to obtain rectangular-like beam. A scanning speed in the direction of liquid transportation was ~1.2 mm/s, whereas the speed of the laser beam returning to a starting point was multiple times higher. Both a high-resolution camera (Canon EOS5D) and stereo microscope (Delta) with a camera (Delta Optics DLT-Cam Pro 5MP with FMA050 fixed microscope adapter) were used to measure the polymer strip deformation and observe liquid transportation.

### Data acquisition methodology

The amount of water transported (pumped) in a large number of scans was measured with the use of an analytical laboratory balance (RAD-WAG AS 60/220.R2), and weight was converted to volume. The number of scans (as well as the time of the scanning process) was dependent on the efficiency of the pumps under the given stimulation. All measurements were made at an ambient temperature of 23 °C. All measurements (for different pump's alignment, width $w$, stimulus length $l$ and maximal intensity $I_m$) were repeated multiple times. Data in the plots correspond to the mean transported volume in a single scan with the error bars corresponding to its standard error. The exact number of used strips, measurements, and scans in each measurement were as below:

Fig. 1e: Four different strips were used for each alignment. Each data point corresponds to five data points, with the average number of scans (from the left): undefined (there was no transport of liquid, as the pump was travelling in the direction opposite to the scanning direction), 89, 60 and 32.

Fig. 3a: Four different strips from the same LCG film were used. Each data point corresponds to five measurements (except $w = 1.6$ mm for which only three measurements were taken due to almost no transport occurring), with the average number of scans (from the left): undefined, undefined (after more than 80 scans no liquid transport was observed), 57, 60, 30, 38, 38, 25 and 25.

Fig. 3d: Five different strips from the same LCG film were used. Each data point corresponds to five measurements, with the average number of scans (from the left): undefined (after more than 40 scans no liquid transport was observed), 49, 22, 21, 16 and 18.

Fig. 4a: Five different strips from the same LCG film were used. Each data point corresponds to five measurements, with the average number of scans (from the left): 32, 29, 26, 25 and 24.

Fig. S3: Each line corresponds to a single sample, with five samples for both directions for the planar alignment and for the homeotropic alignment, and four samples for the splay alignment.

Fig. S4: Both the heating and cooling data points correspond to four separate measurements.

Fig. S6: A single strip cut from an LCG film was measured.

### Numerical simulations

The numerical minimisation was done in Mathematica by creating a list of nodes (~1000) over a domain far greater than the size of the deformation (~4 cm). The curvature was calculated using finite differences and the energy of each node was evaluated using Eq. (1). Minimisation was done using the *NMinimize* routine with the first and the last nodes clamped at zero height, and all the remaining nodes were constrained to remain at or above the surface of the substrate.

## Data availability

The authors declare that all data supporting the findings of this project are available within the paper, its Supplementary information files or on Zenodo repository (DOI: 10.5281/zenodo.7702020).

## Code availability

The authors declare that the Mathematica code for generating the bolus shape for a Gaussian stimulus is available on Zenodo repository (DOI: 10.5281/zenodo.7702020).

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

## Acknowledgements

K.D. was supported by grant 2019/03/X/ST7/02065 ("Ciekłokrystaliczne usieciowane polimery jako elementy wykonawcze w układach mikrofluidycznych") from National Science Centre, Poland (NCN), thanks to which she implemented internship in Smart Photonic Materials Group in the Faculty of Engineering and Natural Sciences at Tampere University, Finland. K.D. and P.W. acknowledge generous funding from the National Science Centre (Poland) within the project 2018/29/B/ST7/00192 "Microscale actuators based on photo-responsive polymers". For the development of this project, M.Z. and Z.D. have received funding from the European Union's Horizon 2020 research and innovation programme under the Marie Skłodowska-Curie grant agreement No 956150. J.B. and K.D. are supported by a UKRI 'future leaders fellowship' grant (grant no. MR/S017186/1). A.P. acknowledges the support from the Academy of Finland, in the framework of PREIN Flagship Programme (Decision No. 320165) and the Center of excellence LIBER (Decision No. 346107). Hao Zeng from Tampere University is acknowledged for fruitful discussions on the topic and for supervising K.D. during her internship at Tampere University.

## Author contributions

K.D.: conceptualisation, design, fabrication, experiments and writing. M.Z.: theory and writing. Z.D.: Characterisation. A.P.: resources and supervision. J.B.: Theory, writing, supervision and funding. P.W.: Conceptualisation, supervision and funding.

## Competing interests

The authors declare no competing interests.
