## [Peer Review File · Nature Communications]

REVIEWER COMMENTS

Reviewer #1 (Remarks to the Author):

In this work, a liquid crystal gel (LCG) strip was prepared by using the splay-bend alignment method. Under the action of medium intensity green light, a bump can be induced in the LCG strip, and a channel is formed between the LCG strip and the substrate. The bump can move with the scanning illumination and thus transport the liquid. The quasi-static analysis explains that the shape of light-driven bump results from the competition between the surface tension of liquid and the bending energy of LCG strip. The critical conditions for liquid transportation are further obtained. This original work is of significance to microfluidic applications, and is recommended to be published after addressing the following comments.

1. It is a neat work. However I feel the novelty has not been well elaborated. Can the authors explain the innovation of the work more explicitly, considering using LCE to manipulate liquid is not new.
2. Different from the preformed channel microfluidic device, the biggest innovation of this work is that the channel position can flexibly change with the scanning light, and the channel can be generated from the interface between LCG strip and rigid substrate. I feel that 1D LCG stripe in current work is insufficient to embody the advantages of this method, and this core idea and advantages need to be further demonstrated, such as constructing a two-dimensional LCG film to realize complex multi-path 2D liquid transportation?
3. Can the authors explain how the fluid inflows and outflows? Do these two processes only involve competition between capillary force and elastic force? Does gravity play a role in outflow and inflow?
4. To verify that gravity is negligible, the illuminated LCG strip should be approximately V-shaped in the absence of liquid, as shown by the schematics in Fig. 2 (a). Can experimental verification be provided?

Reviewer #2 (Remarks to the Author):

The manuscript "Light-driven peristaltic pumping by an actuating splay-bend strip" by Dradrach et al describes trapping and transporting fluids using an optically induced bump. This is an interesting contribution to the field of microfluidics because it dispenses with traditional channels, pumps and connectors. The manuscript is written well following a clear narrative and the theoretical model is accessibly described. In my opinion, the manuscript will be of interest to the readership of Nature Communications.

My only criticism concern the practicalities of implementing the approach:

- the method is well suited to transporting boluses of fluid but not for continuous pumping action. Since every droplet has to be individually guided, the maximum achievable flow ($\mu\text{L}/\text{min}$) will be limited and dependent (among other things) on the length of the channel. Could the authors include a discussion on these limitations?
- It may be interesting to suggest some application scenarios where the unique design can have an advantage over standard microfluidics.

Reviewer #3 (Remarks to the Author):

The manuscript describes a microfluidic pumping method based on the light-driven deformation of a liquid crystalline gel (LCG) strip.

The strip is placed on top of a microfluidic substrate, connecting two wells, with a thin lubrication layer.

The manuscript reports the pumping of the fluid from one well to the other by the peristaltic deformation of the strip induced by a scanning incident laser spot.

The manuscript presents initial experimental results and a mathematical model that describes the pumping mechanism based on the strip deformation.

The manuscript presents only a few experimental results to support its claims, and, most importantly, fails to report important methodological information.

The mathematical modelling is simplistic yet informative, but some aspects of it are unclear or misleading.

Detailed comments follow:

- lines 77-80: LCEs-based systems working in liquids, including water, have been reported in the literature

- line 85: some of the above-mentioned LCEs swim by peristaltic waves or by scanning lasers (the same principle adopted here), even if they were not used for pumping

- lines 114-115 and related methods: a Young's modulus of 40 MPa seems quite high for LCGs; moreover, although it is stated so, the data is not shown in the experimental supplement; also please describe the custom-built system for stress-strain characterization

- line 115 and related supplement and methods: the contraction rate seems small compared to other LC-based materials (100K temperature variation needed for a 10% contraction); also, the contraction data shows a linear trend versus temperature with no evident nematic-to-isotropic transition: why's that? How many samples were tested? How many cycles? Please provide the model number of both the Nikon microscope and the Linkam stage you used

- Fig. 1c-e and related text and methods: how did you measure the transported volume of liquid? Over how many scans? How many samples did you test for each alignment? How did you calculate the averages and errors of transported volumes for the different alignments? I was also surprised that the homeotropic alignment was not considered at all as an option

- line 178-181: using the preferred curvature as a proxy for the intensity of the actuation stimulus is quite unintuitive: as the strains can be expressed as a function of temperature (perhaps by fitting a curve to the reported experimental data), wouldn't it be more intuitive to use the peak temperature in the strip as a model parameter? Moreover, at which temperature was the strain difference calculated? According to the reported data, it should be ~ 0.04 (and not 0.08) at $\sim 50^\circ\text{C}$, which is the working temperature of the LCG strips... how did you get this value?

- line 194: another reference to the supplement about the Young's modulus (this time 30 MPa and not 40 MPa as above); this data is not reported in the supplement

- line 206 and related supplement: x_L and x_R are used, according to my interpretation, as the left and right ends; then the 'R' superscript is used in other variables to mean "rectangular": this is misleading

- rectangular and Gaussian stimuli: a plot of K_R and K_G versus x (in the supplement) will be helpful for the reader

- line 211: what about the variation of surface tension with temperature? what do you mean by "physically reasonable stimulus parameters"? please report them

- Pumping performance section and related movies: the use of K_m as a proxy for the stimulus (light/temperature) intensity is unintuitive; also, the classification of the stimulus (here meant, I guess, as the material response) as weak, intermediate, and strong depending on whether the strip touches the substrate and independently of the intensity and power of the actual (light/temperature) stimulus is very misleading: what about referring to the former as weak, intermediate, and strong "response" or "regime"?

- Fig. 3a and 3d: how were the transported volumes measured? How many experiments/samples? How many scans per sample?

- line 273: why did you choose a K_m of 0.8 mm^{-1} to compare theory and experiments, given that it is stated above that the preferred curvature derived from experiments is instead 1 mm^{-1} ?
- lines 278-285: quite unclear and speculative
- at this point, I was surprised to see no experimental validation of the theoretical optimisation
- line 297: experimentally, how can one assure that the strip is "translationally invariant"
- conclusion: please provide a discussion on how you foresee your pumps to be implemented in microfluidics, considering their potential advantages and limitations

We would like to thank you for your time spent on reviewing our manuscript. We were particularly gratified by reviewer 2's comment that our work is "*written well following a clear narrative and the theoretical model is accessibly described... [and]... of interest to the readership of Nature Communications,*" and also by reviewer 1's comment that our work "*is of significance to microfluidic applications, and is recommended to be published.*" We found the feedback from all the reviewers very helpful, and have made many improvements to the manuscript as a result, including significant new experiments.

In this response, we provide first a summary of the most significant improvements we have made, and then secondly provide detailed answers to each point from the referees. We also supply a copy of the manuscript diff, in which all the improvements are traced with the use of red and blue colours. The SI has also been improved and extended.

Summary of key changes

Complexity

Reviewer 1 and the editor suggested that we should highlight the versatility of our channel-free and light-controlled design, and how it enables more complex actions. We have added a new section "Versatility of the LCG pump" with a new figure (Fig. 5) that demonstrates some possibilities, including reversibility, pumping of complex fluids, strip reconfiguration, and a cross-shaped pump.

Novelty and application

Reviewer 1 and 2 suggested that we could better highlight the novelty of our design relative to previous work on LCEs, and suggest possible areas of application. These themes are now much more prominently discussed in the introduction and conclusion.

Improved experiments

Reviewer 3 asked several questions about precise experimental details, and how theory and experiment are matched. In thinking about these questions, we realised that our choice of a narrow line of laser stimulation (much narrower than the strip width) was unhelpful, as the heated length of the strip was several times longer than the illuminated length (thermal broadening) due to thermal conductivity and the finite rate of cooling post illumination. Although this issue did not impede pumping performance, it introduced a degree of flexibility in fitting theory to experiment, as the experimental stimulation length was actually unknown, and also a degree of confusion as the chosen length was not with the length of the beam. We have now repeated these experiments using a beam that is less intense but much longer, so that thermal broadening is much less significant, and the illuminated length is a good measure of the stimulated length. With this change, we may now compare experiment and theory more clearly and with essentially zero fitting flexibility. The resulting good match (e.g. Fig. 3d) is strong evidence that the theory captures the mechanical essence of the pump.

Optimization experiment

Reviewer 3 was surprised we did not have an experimental verification of the theoretical stimulus-length optimization. We have now done this, as presented in Fig. 4.

Reviewer #1 (Remarks to the Author):

In this work, a liquid crystal gel (LCG) strip was prepared by using the splay-bend alignment method. Under the action of medium intensity green light, a bump can be induced in the LCG strip, and a channel is formed between the LCG strip and the substrate. The bump can move with the scanning illumination and thus transport the liquid. The quasi-static analysis explains that the shape of light-driven bump results from the competition between the surface tension of liquid and the bending energy of LCG strip. The critical conditions for liquid transportation are further obtained. This original work is of significance to microfluidic applications, and is recommended to be published after addressing the following comments.

1. It is a neat work. However I feel the novelty has not been well elaborated. Can the authors explain the innovation of the work more explicitly, considering using LCE to manipulate liquid is not new.

Our response: Although there are numerous publications on using LCEs for locomotion in fluids (i.e. swimming) [29, 37, 52, 53] there are actually remarkably few on using LCEs for fluidic manipulation. The pioneering paper is [26], which embeds a small LCE actuator in a microfluidic channel, and actuates as a valve via embedded Joule heaters. We think this is the only paper using an actual LCE, but there are a few others using different forms of LC actuators. In [18-19] photo-chemically actuated sheets of LCN are embedded in channels to provide pumping and valving in response to UV light, and in [20-21] uncrosslinked LC-polymers (i.e. plastics) are actuated by UV light to deform microfluidic channels, thereby propelling fluid slugs via capillary forces.

Our work differs fundamentally from these five examples because we are not embedding actuating components in/around channels. Instead, we have a channel-free design that produces a host of advantages - it is easy to fabricate, is an open atmospheric-pressure system, and enables reconfigurability/reprogrammability. Furthermore, none of these previous examples uses photo-thermal actuation (a possibility enabled by the development of plasticized LCG, and which avoids damaging UV light) and none use a spatially patterned director to obtain a desired mode of deformation. Of course, photothermal actuation and patterned directors are well established in LCE actuation more generally, but this is an initial exploration of their potential in fluid manipulation. Given these stark differences with previous work, it is perhaps no surprise that the pump also operates via a different and previously unexplored solid-fluid interaction that balances photo-bend and surface tension, and forms a device that is not only new in LCE fluidics, but also a marked departure from all other types of microfluidics.

We have made several changes to the introduction and discussion, which we hope now clarifies these points of novelty.

2. Different from the preformed channel microfluidic device, the biggest innovation of this work is that the channel position can flexibly change with the scanning light, and the channel can be generated from the interface between LCG strip and rigid substrate. I feel that 1D LCG stripe in current work is insufficient to embody the advantages of this method, and this core idea and advantages need to be further

demonstrated, such as constructing a two-dimensional LCG film to realize complex multi-path 2D liquid transportation?

Our response: We agree that our original paper did not show the full potential of the LCG strip system. We have now included a new section exploring the possibilities (“Versatility of the LCG-based peristaltic pump”). In this section we demonstrate peristaltic pump’s reversibility and reconfigurability (Fig. 5a, b and Mov. M6) and pumping of a complex fluid (a slurry - pumpkin soup; Fig. 5c). We have also provided a preliminary demonstration of a more complex system - a cross-like actuator (Fig. 5 d, e) connecting four reservoirs that can pump between any two of them (new Fig. 5 d, e, Mov. M7).

However, although we finish by discussing that a full 2D sheet setup is a natural next step for this system, we feel strongly that it lies beyond the scope of this manuscript, as it forms a fundamentally different system both theoretically and experimentally. Firstly, it would require a high-strain actuating material (as the formation of a fluid-transporting blister is stretch not bend) whereas our LCG is limited to a few percent. A good candidate would be a low-temperature LCE, and these do now exist, but they are made by a very different process. Secondly, 2D would require a pure homeotropic film, which is a different pattern of alignment. Thirdly it would require a more sophisticated optical setup with the spot able to move in 2D rather than one.

Therefore, on an experimental level, 2D requires a different material, different alignment and different optics, and hence is clearly a separate project. Perhaps more importantly, on a conceptual level, the 2D system eliminates bend and surface tension, so it operates via radically different mechanics than described in our paper. Thus our work points towards the 2D system, but is also quite distinct from it.

Overall, we believe that our treatment of the strip geometry is a single self-contained story that identifies and describes a novel active-solid/fluid interaction, and shows how it can be used for pumping. We hope that our additions demonstrating the versatility of the strip geometry suffice for the referee, and look forward to exploring the full 2D system in a later paper, after many months more work!

3. Can the authors explain how the fluid inflows and outflows? Do these two processes only involve competition between capillary force and elastic force? Does gravity play a role in outflow and inflow?

Our response: We agree that it was an oversight to not discuss in/outflow in our original text. We have now included an additional movie showing the processes of water pick-up, transportation and deposition (Mov. M2) and added the following text to the section:

“In our calculations, we have omitted gravity. As always in a fluid, the competition between gravity and surface tension is captured by the capillary length (3 mm for water), with surface tension controlling below this scale, and gravity above. Given the bump amplitude is a magnitude smaller, we indeed expect a surface tension-dominated regime. We confirm this conclusion by numerically including gravity (Fig. S9), and also by observing that the strip can transport a fluid bolus even if the whole apparatus is inverted (Mov. M3). However, these considerations do not apply to the reservoirs, which have diameters comparable to the capillary length, and rely on gravity to contain the fluid. Correspondingly, inflow and outflow

involve elasticity, capillarity and gravity. The strip's end extends over the reservoir and we observe (Mov. M2) that it remains on the water-air interface while it (un)bends during (out)in-flow, and that both processes are effective at various reservoir fill levels. Our interpretation of the motion is that the comparatively large reservoir area reduces capillary forces during in-flow processes, allowing larger amplitude bending that deforms the air-water interface via normal meniscus physics, and can easily capture enough water to fill the bump profile. Similarly, during out-flow, the transported fluid rapidly enters the reservoir upon contact to eliminate the higher surface energy of the sides in favour of a low-energy deformed meniscus.”

4. To verify that gravity is negligible, the illuminated LCG strip should be approximately V-shaped in the absence of liquid, as shown by the schematics in Fig. 2 (a). Can experimental verification be provided?

Our response: We have now added a new Fig. S7, showing the deformation of the strip under local stimulation in air, which is indeed a high amplitude V shape. We actually do not believe this is a good test of gravity being negligible, as the larger weight would be the fluid in the bolus rather than the LCG strip itself. However, as discussed above, we have also now included three lines of evidence that gravity is negligible - the observation that amplitude is considerably less than the capillary length, numerical calculations that include gravity, and a demonstration of bolus transport upside down.

Reviewer #2 (Remarks to the Author):

The manuscript "Light-driven peristaltic pumping by an actuating splay-bend strip" by Dradrach et al describes trapping and transporting fluids using an optically induced bump. This is an interesting contribution to the field of microfluidics because it dispenses with traditional channels, pumps and connectors. The manuscript is written well following a clear narrative and the theoretical model is accessibly described. In my opinion, the manuscript will be of interest to the readership of Nature Communications.

My only criticism concern the practicalities of implementing the approach:

- the method is well suited to transporting boluses of fluid but not for continuous pumping action. Since every droplet has to be individually guided, the maximum achievable flow ($\mu\text{L}/\text{min}$) will be limited and dependent (among other things) on the length of the channel. Could the authors include a discussion on these limitations?

Our response: This is a very interesting point. We have now included a discussion of the achievable flow rate in the updated optimization section, and explicitly noted its dependence on strip length. We have also included a more general discussion of limitations in the last paragraph of the discussion.

We note that the length dependence of the pumping rate assumes that only one bolus is transported at any one time. Although this reflects our demonstration, it would be straightforward for a long strip to transport multiple boluses in series, with multiple light spots, giving a higher and length-independent pumping rate. We have commented on this, along with an estimate of the resultant rate, in our general discussion of limitations.

- It may be interesting to suggest some application scenarios where the unique design can have an advantage over standard microfluidics.

Our response: We have added a new section on versatility, showing some of the unusual capabilities of our channel-less optofluidic system. We also now highlight these advantages and resultant possible use cases in the second paragraph of the discussion.

"[Our] design avoids any need for valves, external mechanical connections, or complex small-scale fabrication. The lack of contact between the fluid and external machinery allows for handling of delicate fluids, potentially whilst sealed in a controlled atmosphere. Furthermore, one may easily modify the strip placement and the pattern of stimulation, both during and between fluidic tasks. Such adaptability could be particularly valuable for rapid prototyping, in-field applications, or where fluidic function must be added post-hoc, e.g. between cell cultures in a Petri dish. More ambitiously, the adaptability points towards a universal "lab-on-a-chip" that, like a full-scale laboratory or a computer chip, can execute many different functions sequentially"

Reviewer #3 (Remarks to the Author):

The manuscript describes a microfluidic pumping method based on the light-driven deformation of a liquid crystalline gel (LCG) strip. The strip is placed on top of a microfluidic substrate, connecting two wells, with a thin lubrication layer. The manuscript reports the pumping of the fluid from one well to the other by the peristaltic deformation of the strip induced by a scanning incident laser spot. The manuscript presents initial experimental results and a mathematical model that describes the pumping mechanism based on the strip deformation.

The manuscript presents only a few experimental results to support its claims, and, most importantly, fails to report important methodological information. The mathematical modelling is simplistic yet informative, but some aspects of it are unclear or misleading.

Our response: Thank you for the valuable and detailed feedback. We have repeated and improved several of our experiments, and now include much more thorough reporting of our methodology. Reassuringly, these improvements have also improved the match between theory and experiment, suggesting that our interpretation of the results is indeed correct.

Detailed comments follow:

- lines 77-80: LCEs-based systems working in liquids, including water, have been reported in the literature

and

- line 85: some of the above-mentioned LCEs swim by peristaltic waves or by scanning lasers (the same principle adopted here), even if they were not used for pumping

Our response: We agree that we should have better highlighted other examples of LCEs working in water, and showing peristaltic undulation. Our revised manuscript now does this in several places:

- (1) In both introduction and discussion, we describe several related references, in particular [18, 19,20, 21, 26], which we believe are the key papers using LC actuators to manipulate fluids. We particularly highlight why our approach is fundamentally different - all these are based on channels, none are photothermal, and none use a patterned director.
- (2) We also highlight [37], which introduced the material we use and showed it is photo-thermally efficacious underwater.
- (3) We now further elaborate peristalsis in the introduction, highlighting several previous works [29,37, 45, 46] that drive peristaltic-like waves of deformation along LCE structures for various forms of locomotion (though none for pumping or fluid manipulation).
- (4) In the revised Discussion we cite several previous works on LCEs for aquatic locomotion [29, 37, 52, 53].

- lines 114-115 and related methods: a Young's modulus of 40 MPa seems quite high for LCGs; moreover, although it is stated so, the data is not shown in the experimental supplement; also please describe the custom-built system for stress-strain characterization

and

- line 194: another reference to the supplement about the Young's modulus (this time 30 MPa and not 40 MPa as above); this data is not reported in the supplement

Our response: Thank you for catching this mistake, the stress-strain data was intended to be in the SI. We have now rectified the omission via Fig. S3. The value of 40 MPa is high for an LCE, but expected for our material, which is a highly crosslinked LCN plasticized by 5CB. It is also the modulus parallel to the director, the perpendicular modulus is substantially lower (8MPa). Our revised manuscript now includes these points.

- line 115 and related supplement and methods: the contraction rate seems small compared to other LC-based materials (100K temperature variation needed for a 10% contraction); also, the contraction data shows a linear trend versus temperature with no evident nematic-to-isotropic transition: why's that? How many samples were tested? How many cycles? Please provide the model number of both the Nikon microscope and the Linkam stage you used

Our response: The comparatively low contraction rate and the lack of a clear phase transition are because our LCG is a densely crosslinked LC-network swollen with 5CB, which acts as a plasticiser softening the network but does not turn it into a lightly crosslinked elastomer (LCE). As with unswollen LCNs, the strain is modest, and the phase transition is suppressed due to a persistent para-nematic phase. The performance of our material is very close to that in [37], which introduced this formulation for actuation in fluids and provided our inspiration. Our introductory text now elaborates on the difference between LCGs and LCEs.

The number of measurements is now written in the "Data acquisition methodology" subsection of Methods. The models of both the Nikon microscope and the Linkam stage are now provided in "Characterization of LCG actuators".

- Fig. 1c-e and related text and methods: how did you measure the transported volume of liquid? Over how many scans? How many samples did you test for each alignment? How did you calculate the averages and errors of transported volumes for the different alignments? I was also surprised that the homeotropic alignment was not considered at all as an option

and

- Fig. 3a and 3d: how were the transported volumes measured? How many experiments/samples? How many scans per sample?

Our response: In “Data acquisition methodology” we now give the model of the used balance, how we have taken all the measurements and the number of measurements and scans for all the data points in the manuscript.

Regarding the homeotropic alignment, we have not previously included it as no transport was observed as the strip was travelling in the direction opposite to the scanning direction. However, we admit it to be an oversight and we have now included it in Fig 1e, in the “Design of the photo-peristaltic pump” subsection, and also in Methods.

- line 178-181: *using the preferred curvature as a proxy for the intensity of the actuation stimulus is quite unintuitive: as the strains can be expressed as a function of temperature (perhaps by fitting a curve to the reported experimental data), wouldn't it be more intuitive to use the peak temperature in the strip as a model parameter? Moreover, at which temperature was the strain difference calculated? According to the reported data, it should be ~0.04 (and not 0.08) at ~50C, which is the working temperature of the LCG strips... how did you get this value?*

and

- *Pumping performance section and related movies: the use of Km as a proxy for the stimulus (light/temperature) intensity is unintuitive; also, the classification of the stimulus (here meant, I guess, as the material response) as weak, intermediate, and strong depending on whether the strip touches the substrate and independently of the intensity and power of the actual (light/temperature) stimulus is very misleading: what about referring to the former as weak, intermediate, and strong "response" or "regime"?*

Our response: We must respectfully disagree that preferred curvature is unintuitive. Preferred curvature is the quantity that appears naturally in the theory, and formulating it in these terms lends the theory a degree of universality, as the same theory would describe any system that actuates with a wave of preferred curvature (e.g. a bi-metal strip, photochemical materials, bi-gel strips etc) whereas the calibration from stimulus to curvature is system specific. However, we do agree that peak temperature is also a very relevant descriptor for our particular experiments. We have therefore directly measured the curvature-temperature relationship for our strip (Fig S5, S6), and then used this data to provide a temperature x-axis for Fig 3b and 3c (in addition to the preferred curvature x-axis) and consistently given temperature figures alongside curvature figures in the text. This direct measurement of the curvature-temperature relation also allows us to bypass the strain-curvature relationship, simplifying and focusing on the narrative.

We agree that our previous terminology of “stimulus regimes” was confusing. We have now renamed them to “response regimes” as it depends on both the stimulus and the strip.

- line 273: *why did you choose a Km of 0.8 mm⁻¹ to compare theory and experiments, given that it is stated above that the preferred curvature derived from experiments is instead 1 mm⁻¹?*

The calculation of 1 mm⁻¹ was intended to be an order-of-magnitude estimate for the appropriate temperature range, rather than a precise value for a given experiment. (As we

wrote, we only wished to show “the theory captures the essence of the pump”). In making figure 2, we thus used the precise value of K_m as a fitting parameter to reproduce the shape seen in the experiment. Interestingly, a slightly higher peak value is required for a Gaussian stimulus vs a rectangle one for the same amplitude of deformation (as the Gaussian decays away from the centre) so a slightly different peak preferred curvature provides the best fit in each case. We now provide an example of each, and, via our new temperature curvature calibration, give the peak temperature used in each case. We also now highlight why they are slightly different, which clarifies that we have used the temperature to fit, although the required value is reassuringly reasonable.

- *line 206 and related supplement: x_L and x_R are used, according to my interpretation, as the left and right ends; then the 'R' superscript is used in other variables to mean "rectangular": this is misleading*

Our response: Thank you for the suggestion. We have now replaced x_L and x_R with x_1 and x_2 to remove ambiguity.

- *rectangular and Gaussian stimuli: a plot of K_R and K_G versus x (in the supplement) will be helpful for the reader - line 211: what about the variation of surface tension with temperature? what do you mean by "physically reasonable stimulus parameters"? please report them*

Our response: Thank you for the good suggestion. We have added a plot of K_r and K_g as Fig. S8.

Surface tension decreases by 15% between 20°C and 80°C, which is small compared to other errors, for example, due to the use of a very simplified model (rectangular). Furthermore, the strip has the highest temperature of the whole system, as the absorption of the light occurs in the strip, and water, due to its contact with the cold substrate and air, has a lower temperature, so the difference in surface tension is even smaller.

The “physically reasonable stimulus parameters” were written in the Figure 2 caption. However, we have now rephrased this paragraph so that the stimulus parameters are also described in this sentence.

- *lines 278-285: quite unclear and speculative*

Our response: We rewrote this paragraph.

- *at this point, I was surprised to see no experimental validation of the theoretical optimisation*

Our response: Thank you for your valuable suggestion. We did the experiment, in which we measured the volume of water transported in a single scan for constant peak light intensity and strip width, but with the varying length of the beam (stimulus). The resulting data is presented in Fig. 4a and described in the “Stimulus length optimisation” subsection.

- *line 297: experimentally, how can one assure that the strip is "translationally invariant"*

Our response: This sentence is no longer in our revised discussion. However, strong evidence that the strip is translationally invariant is now provided by the photos accompanying our temperature-curvature calibration (Fig S6), where one sees that a uniformly heated strip (in a water bath) adopts rolls into an arc of uniform curvature. This directly confirms that the curvature-temperature relation is translationally invariant, and also strongly suggests that the other strip properties (thickness, alignment quality, actuation strain, modulus) are also translationally invariant, as they act in consort to make preferred curvature.

- conclusion: please provide a discussion on how you foresee your pumps to be implemented in microfluidics, considering their potential advantages and limitations

Our response: We have now expanded on this, both via our new section/figure on versatility and also in the second paragraph of the discussion.

REVIEWERS' COMMENTS

Reviewer #1 (Remarks to the Author):

All my concerns have been addressed. I would like to recommend its publication in Nature Communications.

Reviewer #2 (Remarks to the Author):

Issues have been addressed, no further comments

Reviewer #3 (Remarks to the Author):

The authors have fully addressed my previous comments and doubts about the work.

In particular, the revised manuscript provides a clearer description of the work, better contextualisation, the full description of the adopted methodology, a clearer presentation of the theory, and sufficient experimental work to sustain its claims.

For this reason, I recommend publication of this revised version of the manuscript.